# Automated Detection and Differentiation of Stanford Type A and Type B Aortic Dissections in CTA Scans Using Deep Learning

**DOI:** 10.3390/diagnostics15010012

**Published:** 2024-12-25

**Authors:** Hung-Hsien Liu, Chun-Bi Chang, Yi-Sa Chen, Chang-Fu Kuo, Chun-Yu Lin, Cheng-Yu Ma, Li-Jen Wang

**Affiliations:** 1Department of Medical Imaging and Intervention, New Taipei City Municipal Tucheng Hospital, New Taipei City 236043, Taiwan; jerryliu0502@gmail.com (H.-H.L.);; 2Department of Medical Imaging and Intervention, Chang Gung Memorial Hospital, Taoyuan City 333423, Taiwan; cooler@cgmh.org.tw; 3Center for Artificial Intelligence in Medicine, Chang Gung Memorial Hospital, Taoyuan City 333423, Taiwan; zandis@gmail.com; 4Department of Medicine, College of Medicine, Chang Gung University, Taoyuan City 333323, Taiwan; b9002078@cgmh.org.tw; 5Department of Cardiothoracic and Vascular Surgery, New Taipei City Municipal Tucheng Hospital, New Taipei City 236043, Taiwan; 6Department of Artificial Intelligence, Chang Gung University, Taoyuan City 333323, Taiwan; 7Artificial Intelligence Research Center, Chang Gung University, Taoyuan City 333323, Taiwan; 8Division of Rheumatology, Allergy and Immunology, Chang Gung Memorial Hospital, Taoyuan City 333423, Taiwan

**Keywords:** deep learning, aortic dissection, computed tomography

## Abstract

Background/Objectives: To develop and validate a model system using deep learning algorithms for the automatic detection of type A aortic dissection (AD), and differentiate it from normal and type B AD patients. Methods: In this retrospective study, a deep learning model is developed, based on aortic computed tomography angiography (CTA) scans of 498 patients using training, validation and test sets of 398, 50 and 50 patients, respectively. An independent test set of 316 patients is used to validate and evaluate its performance. Results: Our model comprises two components. The first one is an objection detection model, which can identify the aorta from CTA. The second one is a dissection classification model, which can automatically detect the presence of aortic dissection and determine its type based on Stanford classification. Overall, the sensitivity and specificity for Type A AD were 0.969 and 0.982, for Type B AD were 0.946 and 0.996 and for normal cases were 0.988 and 1.000, respectively. The average processing time per CTA scan was 7.9 ± 2.8 s. (mean ± standard deviation). Conclusions: This deep learning automatic model can accurately and quickly detect type A AD patients, and could serve as an imaging triage in an emergency setting and facilitate early intervention and surgery to decrease the mortality rates of type A AD patients.

## 1. Introduction

Aortic dissection (AD) is a life-threatening medical condition, which is primarily due to a tear in the aortic intima, which creates a true and false lumen. The annual incidence of acute AD in the general population is estimated to range from 2.6 to 3.5 per 100,000 people [1,2]. AD can be divided into two categories according to the widely used Stanford classification. The involvement of the ascending aorta is classified as Stanford type A AD, while Stanford type B AD does not involve the ascending aorta. It is clinically important to rapidly distinguish acute Standford type A AD, because it is a cardiovascular surgical emergency. Without surgical intervention, mortality rates are about 1 to 2 percent per hour after symptom onset [3]. The computed tomography angiography (CTA) of the aorta (aortic CTA) can provide detailed assessment of extension of the dissection, and differentiate type A and B AD and thereby guide therapy, which has been reported with a pooled sensitivity of 100% and a specificity of 98% in a meta-analysis [4]. Indeed, aortic CTA has become a crucial imaging tool in emergency patients presented with atypical chest pain and can be served as the first line in diagnosing dissection in today’s medical practice. [5,6]. Nevertheless, despite its critical role, the report turnaround time had a wide range from 0.4 to 628.5 h [7]. The turnaround time could be long due to the complexities of image interpretation and the workload of radiologists, especially in a busy emergency department with a long waiting list. In emergency cases, such as type A AD, if left unnoticed, it could potentially have life-threatening consequences. This highlights the need for developing an automatic algorithm to incorporate clinical workflow, assisting in the early identification of AD.

Deep learning with convolutional neural networks has been shown to succeed in medical imaging segmentation tasks [8], and could be used to develop algorithms for object detections and classifications in images. Deep learning models have been developed to detect AD with a comparable diagnostic performance to radiologists, and showed the potential to support clinical practice [9,10,11]. Raj et al. created an algorithm that could detect AD in the abdomen [12]. In contrast to their study, we intend to develop a deep learning model focusing on AD in the thorax, which could potentially be a very useful tool for automatic detection of the presence of AD for aortic CTA, followed by notifications to clinical physicians and radiologists for confirmation, which facilitates the early initiation of treatment of type A AD patients. Thus, we aimed to develop an automatic artificial intelligence (AI) system using deep learning algorithms as a triage tool to detect and classify AD in this study, in order to reduce type A AD mortality by reducing the time gaps between the end time of aortic CTA examination and the start time of intervention or surgery.

Our proposed system consists of two primary deep learning models aimed at achieving our objectives. The first model is an object detection system designed to locate and crop the aorta area from each slice within a set of CTA images. For this task, we selected RetinaNet [13] due to its effectiveness in handling high class imbalance between the foreground and background instances typical in dense object detection tasks. Introduced by Lin et al., RetinaNet is particularly recognized for its use of focal loss, a specialized loss function that reduces the loss in well-classified examples, allowing the model to focus more on challenging and misclassified instances. This approach enhances the detection capability, especially when dealing with small or infrequent objects. RetinaNet, as a quintessential single-stage detector, integrates the strengths of a feature pyramid network (FPN) within a coherent network structure, contributing to its efficiency in object detection tasks. With its ability to focus on hard-to-detect objects, RetinaNet is highly adaptable and efficient for various datasets and applications, particularly in real-time scenarios that demand both processing speed and prediction accuracy.

The second part of our system is a classification model that identifies whether each aorta patch contains a dissection. For this purpose, we employed EfficientNet-B0, a CNN architecture that maximizes performance relative to resource usage in image classification tasks. Proposed by Tan and Le in 2019 [14], EfficientNet introduces a compound scaling method, which systematically balances and scales depth, width and resolution, thereby improving accuracy while reducing computational costs. Unlike traditional methods that resize dimensions independently, EfficientNet scales all three aspects simultaneously, leading to more effective parameter utilization and an overall improved performance. The EfficientNet-B0 model, a baseline of this architecture, utilizes depth-wise separable convolutions and squeeze-and-excitation blocks. EfficientNet has demonstrated state-of-the-art accuracy in the ImageNet dataset, with a significantly smaller memory footprint and computational cost, marking a crucial advancement in deep learning. This makes EfficientNet particularly advantageous for deployment in resource-constrained environments and diverse applications.

## 2. Materials and Methods

### 2.1. Study Design

This retrospective study was conducted at a single medical center after approval obtained by the Institutional Review Board Committee (IRB) of Chang Gung Memorial Hospital (IRB number: 202100479B0) issued on 29 March 2021. The requirement for informed consent from the patients were waived by the IRB. The dataset collection and experiments were conducted in agreement with the approved ethical guidelines and regulations.

### 2.2. Dataset

From January 2018 to December 2020, there were 2243 patients receiving aortic CTA scans selected from the picture archiving and communication system. Only one exam per patient was admitted to the study to prevent duplicates. A total of 1718 patients met the inclusion criteria, which were as follows: (1) clinical presentation of atypical chest pain, (2) having had an aortic CTA performed and (3) age over 20 years. Four radiologists with at least 5 years of experience reviewed the clinical information and images and excluded the cases based on the following criteria: (1) missing data or medical record (2) suboptimal CTA study showing poor opacification or visualization of aorta, (3) presence of intra-aortic device or surgical change at the time of the study, (4) thrombus in the aorta and (5) other co-existing radiological findings in the thorax, such as pleural effusions, lung mass and consolidation. Figure 1 shows examples of AD and normal cases in our study.

### 2.3. CT Image Acquisition

A non-gated CTA of aorta was performed using 100 milliliters (mL) of iodinated contrast medium (Omnipaque, 350 mg Iodine/mL), followed by 50 mL of saline flush. The arterial phase images were acquired by bolus tracking, with a threshold of 160 Hounsfield units at the ascending aorta. The scan range was set from the lower neck to the groins. The size of all images are 512 × 512 pixels.

### 2.4. Aorta Labeling

Aortic CTA axial images of each patient were analyzed by 4 radiologists for the absence and presence of type A and B AD according to the Stanford classification, which served as ground truth. Each axial image of aortic CTA of chest region of each patient were also labeled by the 4 radiologists as the following: (1) normal ascending aorta, (2) normal aortic arch, (3) normal descending thoracic aorta, (4) dissected ascending aorta, (5) dissected aortic arch and (6) dissected descending thoracic aorta, using a rectangular bounding box for each. The window level was set to a center of 40 and width of 400 Hounsfield units. The ascending aorta began at the level of aortic valve while the descending thoracic aorta ended at the last axial image of the lung parenchyma which remained present. Joint meetings reviewing the labeling were held prior to further analysis.

### 2.5. Main Framework

The code that was used to run the model is available at: https://github.com/brainma/aortic_dissection_det_pytorch/tree/main (accessed on 18 September 2024).

In our system (Figure 2), the first step used an object detection model to identify three parts of the aorta—namely, the ascending aorta, aortic arch and descending aorta—from the axial view of a CTA scan. Subsequently, we extracted and cropped each detected aorta into an individual image. These cropped patches were then resized into 100 × 100 pixels and sent to a classification model to determine the presence or absence of a dissection in each slice. After classification, the aortic patches were arranged in a linear sequence reflecting their anatomical structure, beginning with the ascending aorta, followed by the aortic arch and ending with the descending aorta. A sliding window analysis, encompassing a span of seven consecutive images, was employed to scrutinize the sequence. If dissection was detected in at least five of the seven images with involvement of the ascending aorta, the system would classify the condition as type A AD and with involvement of the ascending aorta, type B AD would be categorized. If dissection was not detected in at least five of the seven images, the patient’s case would be considered normal. For the objection detection model, we utilized a RetinaNet [13] architecture with a ResNet50 [15] encoder as its backbone. Optimization was performed using the Adam algorithm with an initial learning rate of 0.00001, and training was conducted in batches of 72. For the classification model, an EfficientNet B0 [14] architecture was employed. The Adam optimization algorithm is again utilized, this time with a learning rate of 0.001, and the model processes data in batches of 512.

### 2.6. Statistical Analysis

For the objection detection model, the precision and recall of each of the 3 aorta locations (ascending aorta, aortic arch and descending aorta) were analyzed using the labeling of 4 radiologists as the ground truths. For the dissection classification model, precision, recall and accuracy, as well as false positive and negative rates, were analyzed in addition to the F1 score and area under a receiver operating characteristics curve (ROC). For the diagnostic performance of the whole algorithm, including detection and dissection models, the sensitivity, specificity, positive predict value and negative predict value of Stanford type A AD, Stanford type B AD and the normal were calculated and demonstrated by the confusion matrix, using the radiologists’ diagnosis for each patient as ground truths. A two-sided *p* value less than 0.05 was considered statistically significant, and a 95% confidence interval was calculated. All statistical analyses were performed using SPSS v26 for Windows.

## 3. Result

### 3.1. Patient Selection and Demographic

From January 2018 to December 2020, 2243 patients receiving aortic CTA scans A total of 1718 patients met the inclusion criteria, and 903 patients were excluded. In total, 814 different patients were enrolled in this study. Among them, 86 patients were diagnosed with Stanford type A AD, 59 patients were diagnosed with Stanford type B AD and 669 patients were normal. A total of 498 patients were selected for the development of the model, who were further divided into the following three sets randomly: training set, validation set, and testing set for the training object detection model (Table 1). From the same 498 patients, we cropped all the labeled aorta areas into 31,622 patches which were further randomly divided into the training set, validation set, and testing set for the training classification model (Table 2). A figure of 316 patients constituted an independent external test set. Figure 3 shows the process of participants selection. The patient characteristics of the developing model dataset and the independent external test set are detailed in Table 3. There are no significant differences in age and gender between the two datasets.

### 3.2. Objection Detection Model

The efficacy of our aorta detection model was as follows: the mAP is 0.938, and the precision values for the ascending aorta, aortic arch and descending aorta were 0.906, 0.975 and 0.977, respectively, while the recall rates for these regions were 0.925, 0.928 and 0.981, respectively.

### 3.3. Dissection Classification Model

As for the dissection classification model, its performance metrics on the test set showed an accuracy of 0.996. The model demonstrated a false negative rate of 0.004 and a false positive rate of 0.005. Its precision and recall (sensitivity) both matched 0.996. The F1 score aligned with these metrics at 0.996, while the specificity was slightly lower at 0.995, and the area under ROC reached the maximum value of 0.990.

### 3.4. Whole System Evaluation

Our aortic dissection screening system was further evaluated on an independent test set comprising 316 patients, of which 247 were normal cases, 32 had Type A AD, and 37 had Type B AD. For normal cases, the system achieved a sensitivity of 0.988 and a specificity of 1.000. In detecting Type A AD, the sensitivity was 0.969 with a specificity of 0.982, while for Type B AD, the sensitivity and specificity were 0.946 and 0.996, respectively. The results were consolidated in a confusion matrix presented in Figure 4. The average processing time per aortic CTA study was 7.9 ± 2.8 s (mean ± standard deviation).

## 4. Discussion

We developed a deep learning-based model that could successfully detect AD and determine its Stanford type. The primary clinical purpose of this model was to serve as a triage tool, with the intention of assisting and flagging potential cases for radiology review. In scenarios where radiology expertise was unavailable or delayed, the model could provide value by highlighting critical imaging findings for clinicians. As a result, the algorithm needed to be accurate in identifying and classifying AD with a fast-processing time. The accuracy of our model was 0.981, and therefore, it possibly could reach human diagnostic ability. The average processing time was 7.9 s per case, which would be timesaving for clinicians. Wang [16] developed a commercialized automatic product that could both detect AD and intramural hematoma (IMH), and its process time was 37.9 s per study. Nonetheless, the detection of AD and IMH together using AI algorithms would increase many notifications due to IMH detection, which is not an emergency situation. Howard et al. [17] reported that 48.6% of type A AD patients die prior to hospital assessment, and the mortality rate of type A AD has been reported as 0.5% to 1.0% per hour in previous studies [3,18]. An early and prompt diagnosis is crucial, since type A AD could be treated by surgery, thus lowering the mortality rate [18]. Aortic CTA served as the most important diagnostic tool for AD. However, the average final report turnaround time differs. Even in the presence of radiologists dedicated to reports from emergency departments, the median final report turnaround time still took 2.75 h [7]. Our model had a very fast processing time, and could be a triage tool to alert clinicians and radiologists to identify this emergency as soon as possible.

Segmentation is an important step to develop an AD diagnostic model [19]. Cao et al. [20] created an automated type B AD segmentation by manually segmenting the true and false lumen of aorta, which is a time-consuming task. Our model used a rectangular labeling box for aorta detection instead of segmentation, which eased the workload of physicians during model development. Thus, we were able to enroll more than 800 patients in this study. Besides that, various models [21,22,23] were developed for segmentation and aortic diameter measurement. However, these studies mainly focused on type B AD. Whether the ascending thoracic aorta was involved in AD or not was clinically crucial, because type A AD is a surgical emergency. A two-dimensional (2D) framework was applied in our model so we could simplify the imaging process and lower the demand for computing power. Hata and Yi et al. [9,10] developed aortic dissection detection models with non-enhanced CT studies. Previous studies [10,24,25] often employed a three-dimensional framework, and we demonstrated comparable sensitivity and specificity as shown in Table 4 using a 2D model.

To reduce false negatives in detecting type A AD, we employed a sliding window analysis method. Hata et al. [9] proposed a consecutive slices algorithm for AD detection, and they concluded that five consecutive slices may be appropriate in the clinical setting. Huang et al. [24] also used five slices in their AD detection model. Reviewing the original CT images, we discovered that all false negatives, including three normal cases being misinterpreted as type A AD, were due to a streaking artifact, which was caused by a dense contrast medium in the superior vena cava [26]. The highly attenuated contrast medium may produce a metal-like appearance in the CT scans due to the combination of a beam hardening-artifact, photon starvations and a scatter artifact [27]. These were commonly encountered artifacts because an optimal CTA of the aorta requires a concentrated bolus of contrast medium, injected from venous access in the upper extremities. The artifacts could be minimized by using filtration, calibration correction and beam-hardening correction software [26]. Experienced physicians were generally able to distinguish between artifacts and true intima flaps. One Stanford type A AD case was considered as type B AD by the model. The entry site of intimal flap in this case was in the distal ascending aorta, in proximity to the aortic arch. Further investigations and larger data sets for training may be needed in the future.

This study has several limitations. First, the abdominal aorta was not included, although detecting the full extent of AD using the aortic diameter measurement may guide the treatment strategy for the patient. The exclusion of abdominal aorta in this study was based on the aim of our algorithms for detecting AD involvements in the ascending aorta, Second, thrombus in the aorta was excluded during model training. Thrombosis was usually observed in the chronic stage of AD. Our primary goal was to develop a model for acute AD detection instead of follow-up care. Besides that, thrombus within the aorta could obscure or compromise the visualization of the intimal flap. As a result, we excluded these patients during data selection. Future work may be required involving additional training to address cases with thrombus to enhance our model’s clinical applicability. Third, the imaging sets were enrolled from a single institute using CT scanners from several vendors. Also, the imaging protocols for aortic CTA may vary in different hospitals. Further studies of the performance of this AI model system in many institutes will be conducted for further assessment and to increase its generalizability.

The development of a deep learning model capable of automatically detecting aortic dissection and classifying it based on the Stanford classification may have an implication in clinical practice, since early and accurate diagnosis is critical in managing AD. We expected to incorporate our model into the workflow of daily practice to serve as a triage tool to assist both clinicians and radiologists. The system could aid in prioritizing surgical intervention for type A AD, for which urgent treatment is required. Our model was able to determine AD. Future research focusing on enhancing our model may be performed by identifying other acute aortic syndromes, including intramural hematoma and penetrating atherosclerotic ulcers. It is also important to expand our model to incorporate with other imaging modalities, since multimodality imaging, such as aortic CTA, echocardiography and magnetic resonance imaging, has been shown to play an important role in the diagnosis, complications and clinical management of AD [28].

## 5. Conclusions

This study demonstrates that AD on CTA could be automatically detected by deep learning algorithms with a high accuracy (0.981), with classifications of AD types provided based on Stanford classification. This automatic triage system takes less than 10 s for inferences of AD and its type for each aortic CTA examination. The implementation of these algorithms could potentially accelerate clinical workflow, by enabling rapid and precise diagnosis, which is crucial for conditions that require urgent surgical intervention, such as type A AD. The triage system could provide real-time alerts for clinicians and radiologists once aortic CTA is performed. Future research for the further validation of the model across different healthcare systems may be helpful for managing AD.

## Figures and Tables

**Figure 1 diagnostics-15-00012-f001:**
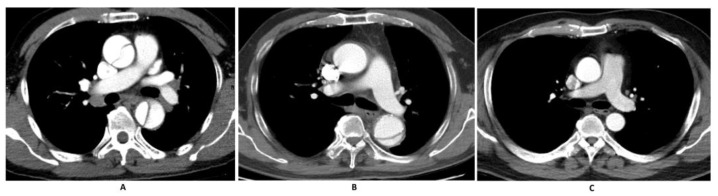
Example of aortic CTA in our study. (**A**) showed aortic dissection, and (AD) involved the ascending and descending aorta, which was the typical imaging presentation of Stanford type A AD. (**B**) displayed an intimal flap in the descending aorta, suggesting a Stanford type B AD. (**C**) was a normal case.

**Figure 2 diagnostics-15-00012-f002:**
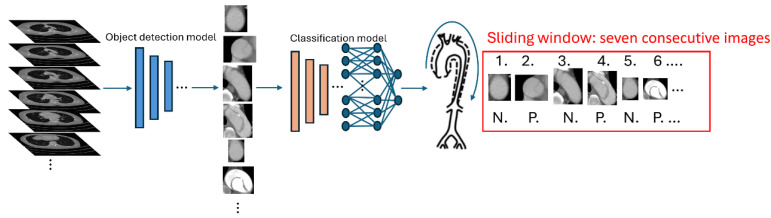
Diagram shows deep learning algorithms. An objection detection model would identify the ascending aorta, aortic arch, and descending aorta from CTA scan. These cropped patches were then sent to the classification model to determine the presence of aortic dissection. After classification, the cropped patches were arranged in a sequence from ascending to descending aorta to determine their Stanford type. The “P.” and “N.” labels under each cropped patch indicate whether the patch is positive or negative for the presence of dissection.

**Figure 3 diagnostics-15-00012-f003:**
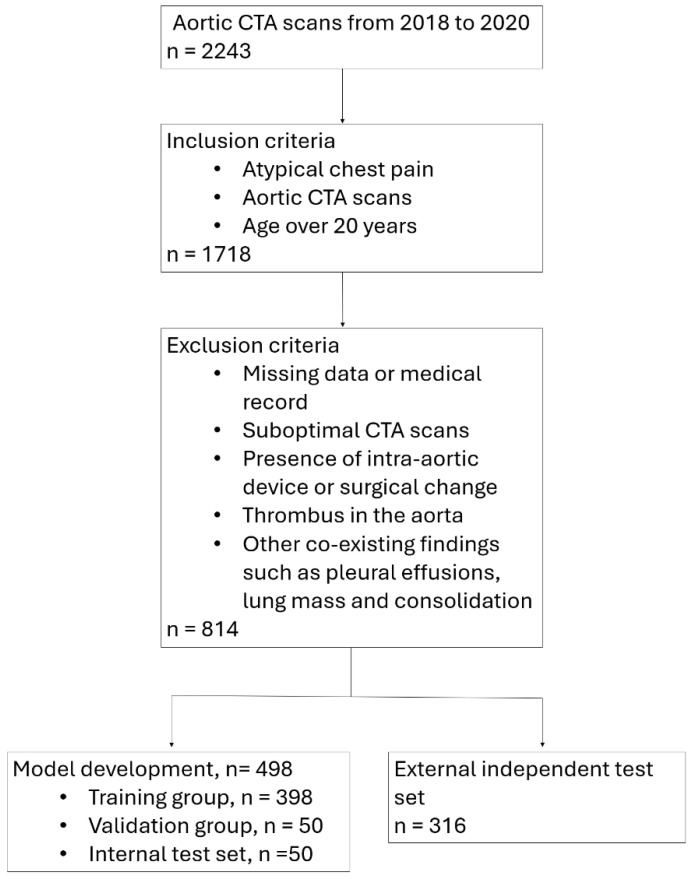
Flowchart shows participants selection. Among the 2243 patients with aortic CTA scans from 2018 to 2020, 1718 patients were included if the following criteria was met: (1) clinical presentation of atypical chest pain, (2) an aortic CTA for evaluating aortic dissection was performed and (3) age over 20 years. A total of 903 patients were excluded because of (1) missing data or medical record, (2) suboptimal CTA scans, (3) presence of intra-aortic device or surgical change, (4) thrombus in the aorta and (5) other co-existing findings. A total of 814 patients were enrolled. A figure of 498 patients were randomly assigned for model development, and 316 patients constitute the independent test set.

**Figure 4 diagnostics-15-00012-f004:**
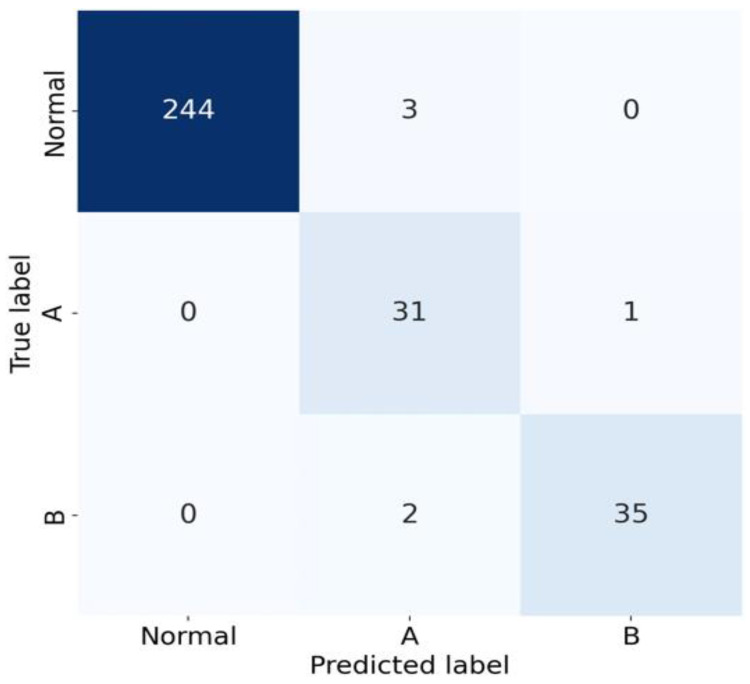
The 3 × 3 confusion matrix. The case numbers of the whole system’s prediction and imaging diagnoses, obtained from the independent test set, were demonstrated in the 3 × 3 confusion matrix.

**Table 1 diagnostics-15-00012-t001:** Datasets for development of the models.

Set	Ascending Aorta	Aortic Arch	Descending Aorta	Images
Training	7191	2709	19,334	21,992 (398 patients)
Validation	927	343	2575	2916 (50 patients)
Test	885	353	2433	2787 (50 patients)

**Table 2 diagnostics-15-00012-t002:** Datasets for development of the classification model.

Set	Positive	Negative	Images
Training	1516	25,765	27,281
Validation	303	35,797	3882
Test	262	197	459

**Table 3 diagnostics-15-00012-t003:** Characteristics of datasets.

	Model Developing Dataset	Independent Testing Dataset	*p* Value
Age (year)	59.30 ± 13.51	60.32 ± 13.57	0.296
Sex			0.058
Male	277 (55.6%)	197 (62.1%)	
Female	221 (44.4%)	119 (37.7%)	
Aortic dissection			<0.001
Normal	422 (84.7%)	247 (78.2%)	
Type A	54 (10.8%)	32 (10.1%)	
Type B	22 (4.4%)	37 (11.7%)	

**Table 4 diagnostics-15-00012-t004:** Comparison of different deep learning algorithms for aortic dissection detection. AD: aortic dissection.

Study [Reference], Year, Country	Sample Size	Framework	Imaging Modality	Dissection Type Interpretation	Intramural Hematoma Identification	Sensitivity (%)	Specificity (%)
Our model 2024, Taiwan	814 (145 AD)	2D	CTA	Yes	No	96.9 (Type A) 94.6 (Type B) 98.8 (Normal)	98.2 (Type A) 99.6 (Type B) 100.0 (Normal)
Hata et al. [9] 2020, Japan	170 (85 AD)	2D	Non-enhanced CT	No	No	90.0	88.2
Yi et al. [10] 2022, China	452 (185 AD)	3D	Non-enhanced CT	Yes	No	86.2	92.3
Huang et al. [24] 2022, Taiwan	130 (100 AD)	3D	CTA	Yes	No	95.45 (Type A) 79.31 (Type B) 93.53 (Normal)	98.55 (Type A) 94.05 (Type B) 94.12 (Normal)
Harris et al. [25] 2019, USA	34,689 (112 AD)	3D	CTA	No	No	87.8	96
Wang [16] 2024, USA	380 (156 AD)	No detail	CTA	No	Yes	92.9	91.5

## Data Availability

The code that was used to run the model is available at: https://github.com/brainma/aortic_dissection_det_pytorch/tree/main (accessed on 18 September 2024).

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
