# Peer review of "Automated Detection and Differentiation of Stanford Type A and Type B Aortic Dissections in CTA Scans Using Deep Learning"

_diagnostics, 2024, doi:10.3390/diagnostics15010012_

Round 1

Reviewer 1 Report

Comments and Suggestions for Authors

The manuscript presents an automated, AI-based model for the rapid and accurate detection and classification of Stanford type A and B aortic dissections in CT angiography. The big question, as with many AI studies, is where the clinical need is when a radiologist is on site. Please show me a radiologist who would miss an aortic dissection in an emergency setting. It doesn't really happen. Therefore, the clinical impact of the study is actually quite limited because such an algorithm would only be considered in institutions where there is no radiologist on site. But is that the direction we want to go in? Furthermore, the results of the study are poor in generalisability because only data from a single institution was used. How can this be transferred to other institutions and settings? While the manuscript compares the proposed method with other models, sufficient details regarding the data set or key performance indicators are missing for some comparisons.

Furthermore, the abdominal aorta and cases with thrombi were excluded, which further limits the applicability. How am I supposed to know in advance whether the patient has a thrombus or not? In other words, a radiologist is needed again to look at the images, and he can actually tell just as well whether an aortic dissection is present or not. So what is the point of such an algorithm? It's nice to show that it works, but for what clinical purpose? All these points should be discussed in detail. Furthermore, is there any robust solution for dealing with contrast-induced streak artefacts?

Introduction: Perhaps go into studies with deep learning for cardiovascular imaging in more detail

Methods:

- Patient selection -> what was the reason for excluding patients with thrombus

Results: Clearly described and well demonstrated. 

Discussion: As mentioned above an outline of specific steps and application scenarios for clinical use are necessary. Why do we need this algorithm? Would you use it in an emergency setting with with thoracic surgeons standing behind you when you are reviewing the images. or does it have the potential for the thoracic surgeons to hijack the tool without involving a radiologist? Please discuss considerations regarding official approval. 

Figures: The image quality could be improved by adding further annotations to the images. The study flowchart could be formatted more clearly.

Author Response

Comments 1:

The manuscript presents an automated, AI-based model for the rapid and accurate detection and classification of Stanford type A and B aortic dissections in CT angiography. The big question, as with many AI studies, is where the clinical need is when a radiologist is on site. Please show me a radiologist who would miss an aortic dissection in an emergency setting. It doesn't really happen. Therefore, the clinical impact of the study is actually quite limited because such an algorithm would only be considered in institutions where there is no radiologist on site. But is that the direction we want to go in? Furthermore, the results of the study are poor in generalisability because only data from a single institution was used. How can this be transferred to other institutions and settings? While the manuscript compares the proposed method with other models, sufficient details regarding the data set or key performance indicators are missing for some comparisons.

Response 1:

Thank you for your constructive feedback. The authors acknowledged that diagnosing aortic dissection via computed tomography angiography can be very straightforward most of the time. Therefore, the primary clinical purpose of this algorithm was to serve as a triage tool. In scenarios where radiology expertise was unavailable or delayed, the algorithm could automatically provide value by highlighting critical findings. If aortic dissections were identified in a small institution, transferring the patients to a tertiary hospital could be done as soon as possible. Revised manuscript regarding the clinical application was added in the first paragraph of discussion. The changes could be seen on page 7, line 7-12.

Our model was developed from datasets in a single center. The authors acknowledged that this condition might have an impact on reproducibility. Our hospital was a tertiary center so that most of patients with acute aortic dissection would be transferred to. We were also one of the largest medical systems in Taiwan with more than 10 hospitals in the group, sharing similar imaging protocols in aortic CTA. We believed that our model had an opportunity to be able to reproduce its capabilities and further work needed to be performed.

Comments 2:

Furthermore, the abdominal aorta and cases with thrombi were excluded, which further limits the applicability. How am I supposed to know in advance whether the patient has a thrombus or not? In other words, a radiologist is needed again to look at the images, and he can actually tell just as well whether an aortic dissection is present or not. So what is the point of such an algorithm? It's nice to show that it works, but for what clinical purpose? All these points should be discussed in detail. Furthermore, is there any robust solution for dealing with contrast-induced streak artefacts?

Response 2:

Thank you for your kind feedback. The authors acknowledged that excluding the abdominal aorta and cases with thrombi may limit the algorithm's applicability. We aimed to focus on the thoracic aorta because Stanford classification was based on presence of intimal flap in the ascending aorta, which was the primary reason for excluding the abdominal aorta. As for thrombosis in the aorta, which was usually observed in the chronic stage of aortic dissection. Our intention was to develop a model for acute aortic dissection detection instead of follow-up care. Besides that, thrombus within the aorta could obscure or compromise the visualization of the intimal flap. As a result, we excluded these patients during data selection. However, we understood that future work may be required involving additional training to address cases with thrombus and abdominal aorta to further enhance our model’s clinical applicability. Lastly, we revised the manuscript in the discussion panel as follows: Thrombosis was usually observed in the chronic stage of AD. Our primary goal was to develop a model for acute AD detection instead of follow-up care. Besides that, thrombus within the aorta could obscure or compromise the visualization of the intimal flap. As a result, we excluded these patients during data selection. Future work may be required involving additional training to address cases with thrombus to enhance our model’s clinical applicability. Changes can be seen on page 9, line 16-21.

Comments 3:

Introduction: Perhaps go into studies with deep learning for cardiovascular imaging in more detail

Methods: Patient selection -> what was the reason for excluding patients with thrombus

Results: Clearly described and well demonstrated.

Discussion: As mentioned above an outline of specific steps and application scenarios for clinical use are necessary. Why do we need this algorithm? Would you use it in an emergency setting with with thoracic surgeons standing behind you when you are reviewing the images. or does it have the potential for the thoracic surgeons to hijack the tool without involving a radiologist? Please discuss considerations regarding official approval.

Figures: The image quality could be improved by adding further annotations to the images. The study flowchart could be formatted more clearly.

Response 3:

Thank you for your kind comments. We have revised the manuscript in the introduction section addressing the issues of deep learning in the imaging of aortic dissection: Deep learning models have been developed to detect AD with comparable diagnostic performance to radiologists, which showed the potential to support clinical practice (9-11). Raj et al created an algorithm that could detect AD in the abdomen (12). In contrast to their study, we intended to develop a deep learning model focusing on AD in the thorax, that would potentially be a very useful tool for automatic detection of the presence of AD for aortic CTA… The changes could be found on page 2, line 16-20.

The rationale of excluding thrombi in the aorta was added in the manuscript as discussed in previous comment, which could be found in the revised manuscript on page , line 16-21.

The primary goal was to develop an automatic aortic dissection detection to serve as a triage tool. We’ve revised the manuscript to further elaboration and as follows: The primary clinical purpose of this model was to serve as a triage tool, with the intention of assisting and flagging potential cases for radiology review. In scenarios where radiology expertise was unavailable or delayed, the model could provide value by highlighting critical imaging findings for the clinicians. As a result, the algorithm needed to be accurate in identifying and classifying AD with a fast-processing time. Changes can be found on page 7, line 7-12.

Reviewer 2 Report

Comments and Suggestions for Authors

The authors conducted a study testing the performance of the methods with different performance measurement techniques using deep learning techniques in the detection and classification of Stanford type a and b aortic dissections in CTA images.When the study is examined, it will be beneficial for the readers to better understand the article by eliminating the issues mentioned below.

1-The study is quite impressive in terms of motivation. Detection and classification of aortic dissection types a and b with the help of deep learning methods, a sub-branch of artificial intelligence techniques, will provide an opportunity for early diagnosis of the patient and therefore early treatment of the disease.

2-It was noticed that the literature review regarding the study was not done sufficiently. All studies on the problem for the years 2023 and 2024 should be carefully examined.

3-In the introduction section of the article, information about the proposed deep method should be given. Such as which deep models are used or whether feature selection is made.

4-Authors should include the study's ethical report information in the materials and methods section of the article. 

5-The authors stated that "If dissection was detected in at least five of the seven images with involvement of the ascending aorta, the system would classify the condition as type A AD and with involvement of the ascending aorta, type B AD would be categorized." Why is the presence of 5 dissections important for classification?

6-It is mentioned that RetinaN method is used for object detection and Efficientnetb0 architecture is used for classification. However, no information is given about how segmentation is performed. No information is given about the images obtained after the segmentation process. Detailed information should be given about the image features obtained after the methods and process steps. The representation of this information with shapes is also extremely important.

7-The proposed model roughly includes object detection and classification stages. However, since no details are given, this figure should be deleted and a new figure should be drawn that reveals the model in detail.

8-The authors used the sentence "Statistical Product and Service Solutions version 26 for Windows." Instead, they should prefer to write SPSS v26.

9-Why did the authors use only the Efficienetnetb0 architecture? Was the softmax activation function chosen as the classifier? Why were machine learning classifiers such as SVM, KNN, and Decision Tree never used in the experiments?

10-The engineering side of the study, namely the segmentation operations, object detection, and especially other parts are extremely unclear. There is not enough information about the techniques and architectures used. No clear information is given about the proposed model and the techniques used. The results obtained by these models and techniques are not fully given. These parts need to be clarified.

11-Please explain why you chose the RetinaN model.

12-"Sliding window analysis model" What is the feature of this model? How does it work? What are its hyperparameters? Detailed information should be given. What is the flow chart like?

13-The conclusion of the study should be expanded. The achieved success should be emphasized.

Author Response

Author's Reply to the Review Report (Reviewer 2)

Comment 1: The study is quite impressive in terms of motivation. Detection and classification of aortic dissection types a and b with the help of deep learning methods, a sub-branch of artificial intelligence techniques, will provide an opportunity for early diagnosis of the patient and therefore early treatment of the disease.

Response 1: Thank you for your kind comment.

Comment 2: It was noticed that the literature review regarding the study was not done sufficiently. All studies on the problem for the years 2023 and 2024 should be carefully examined.

Response 2: Thank you for your kind suggestions. We’ve revised the manuscript in introduction section as recommended. The changes were as follows and could also be found on page 2, line 16-20. Deep learning models have been developed to detect AD with comparable diagnostic performance to radiologists, which showed the potential to support clinical practice (9-11). Raj et al created an algorithm that could detect AD in the abdomen (12). In contrast to their study, we intended to develop a deep learning model focusing on AD in the thorax, that would potentially be a very useful tool for automatic detection of the presence of AD for aortic CTA…

Comment 3: In the introduction section of the article, information about the proposed deep method should be given. Such as which deep models are used or whether feature selection is made.

Response 3: Thank you for your helpful suggestions; we have added two paragraphs to the final part of the introduction section on page 2 to address this.

Comment 4: Authors should include the study's ethical report information in the materials and methods section of the article.

Response 4: Thank you for your kind suggestions. The ethical report information was added to the materials and methods section. You can see the changes on section 2.1 in page 3.

Comment 5: The authors stated that "If dissection was detected in at least five of the seven images with involvement of the ascending aorta, the system would classify the condition as type A AD and with involvement of the ascending aorta, type B AD would be categorized." Why is the presence of 5 dissections important for classification?

Response 5: Thank you for your question. We use the presence of dissections in five out of seven consecutive slices as a criterion to confirm the occurrence of a dissection and reduce false positives. Several previous studies listed immediate below have also employed five consecutive dissections as the threshold for a similar mechanism.

1)  Hata, A.; Yanagawa, M.; Yamagata, K.; Suzuki, Y.; Kido, S.; Kawata, A.; Doi, S.; Yoshida, Y.; Miyata, T.; Tsubamoto, M.; et al. Deep learning algorithm for detection of aortic dissection on non-contrast-enhanced CT. Eur Radiol 2021, 31 (2), 1151-1159. DOI: 10.1007/s00330-020-07213-w  From NLM Medline.

2)  Huang LT, Tsai YS, Liou CF, Lee TH, Kuo PP, Huang HS, Wang CK. Automated Stanford classification of aortic dissection using a 2-step hierarchical neural network at computed tomography angiography. Eur Radiol. 2022;32(4):2277-85.

Comment 6: It is mentioned that RetinaN method is used for object detection and Efficientnetb0 architecture is used for classification. However, no information is given about how segmentation is performed. No information is given about the images obtained after the segmentation process. Detailed information should be given about the image features obtained after the methods and process steps. The representation of this information with shapes is also extremely important.

Response 6:

Thank you for your question. In fact, our system does not include a semantic segmentation step. We apologize for any confusion our description may have caused. We have revised our descriptions in sections 2.3, 2.5 on page 3 and 4, and the caption of Figure2 on page 4 for clarity.

Comment 7: The proposed model roughly includes object detection and classification stages. However, since no details are given, this figure should be deleted and a new figure should be drawn that reveals the model in detail.

Thank you for your suggestion. We have revised our descriptions in in sections 2.3, 2.5 on page 3 and 4, and the caption of Figure2 on page 4 for clarity.

Comment 8: The authors used the sentence "Statistical Product and Service Solutions version 26 for Windows." Instead, they should prefer to write SPSS v26.

Response 8:

Thank you for your kind feedback, we’ve revised the sentence as suggested. This change could be found on page 5, line 2.

Comment 9: Why did the authors use only the Efficienetnetb0 architecture? Was the softmax activation function chosen as the classifier? Why were machine learning classifiers such as SVM, KNN, and Decision Tree never used in the experiments?

Response 9:

Thank you for your question. Based on a lightweight architecture, we selected EfficientNet B0 to act as our classifier as inference speed was very important for our use-case. The last layer of our classifier is a fully connected layer and not a softmax activation function. The traditional machine learning algorithms like SVM, KNN, and Decision Trees were not chosen because those methods do not extract significant features on their own and we need to feature engineer. One of the most important advantages of deep learning and neural networks is the automatic identification of the relevant features of the data. On the other hand, handcrafted features are usually not only shallow representations and based on statistical methods which may fail to capture meaningful features with deeper information. Machine learning methods generally need large datasets for the system performance improvement, so this limitation can be caused to lead to less accuracy. This is where the deep learning approaches comes handy. Previous studies have shown deep learning and neural network approaches outperform other methods for image classification problems (1-3).

1)     Yanzheng Yu, Deep Learning Approaches for Image Classification, EITCE '22: Proceedings of the 2022 6th International Conference on Electronic Information Technology and Computer Engineering, pp 1494 - 1498, DOI: https://doi.org/10.1145/3573428.3573691

2)       Yunfei Lai, A Comparison of Traditional Machine Learning and Deep Learning in Image Recognition, ICEMCE 2019, DOI: doi:10.1088/1742-6596/1314/1/012148

3)       Alex Krizhevsky, Ilya Sutskever, and Geoffrey E. Hinton, ImageNet Classification with Deep Convolutional Neural Networks. NeurIPS 2012

Comment 10: The engineering side of the study, namely the segmentation operations, object detection, and especially other parts are extremely unclear. There is not enough information about the techniques and architectures used. No clear information is given about the proposed model and the techniques used. The results obtained by these models and techniques are not fully given. These parts need to be clarified.

Response 10:

Thank you for your suggestion. We have revised our descriptions in sections 2.3, 2.5 on page 3 and 4, and the caption of Figure2 on page 4 for clarity.

Comment 11: Please explain why you chose the RetinaN model.

Response 11:

Thank you for your question. We selected RetinaNet as our object detection model due to its use of Focal Loss, a specialized loss function that diminishes the loss for well-classified examples. This enables the model to concentrate more on challenging and misclassified instances, thereby enhancing its detection capabilities, particularly for small or infrequent objects. RetinaNet is a quintessential single-stage detector that integrates the strengths of a feature pyramid network (FPN) within a cohesive architecture, which contributes to its efficiency in object detection tasks. Its ability to focus on hard-to-detect objects makes RetinaNet highly adaptable and efficient across various datasets and applications, especially in real-time scenarios that require both processing speed and prediction accuracy. For our purpose, we need both accuracy and inference speed. We have also added this description to the second-to-last part of the introduction section on page 2.

Comment 12: "Sliding window analysis model" What is the feature of this model? How does it work? What are its hyperparameters? Detailed information should be given. What is the flow chart like?

Response 12:

Thank you for bringing this to our attention. Sliding window analysis is not a machine learning model; rather, it is a rule-based process designed to reduce false positives in predictions by leveraging the observed fact that dissections typically occur consecutively. Several previous studies have employed similar mechanisms to enhance their predictions (1,2).

1) Hata A, Yanagawa M, Yamagata K, Suzuki Y, Kido S, Kawata A, et al. Deep learning algorithm for detection of aortic dissection on non-contrast-enhanced CT. Eur Radiol. 2021;31(2):1151-9.

2). Huang LT, Tsai YS, Liou CF, Lee TH, Kuo PP, Huang HS, Wang CK. Automated Stanford classification of aortic dissection using a 2-step hierarchical neural network at computed tomography angiography. Eur Radiol. 2022;32(4):2277-85.

Comment 13: The conclusion of the study should be expanded. The achieved success should be emphasized.

Response 13:

Thank you for your kind suggestion. We’ve revised the manuscript in the conclusion section to elaborate the potential clinical applications as follow: The implementation of the algorithms could potentially accelerate clinical workflow by enabling rapid and precise diagnosis, which was crucial for conditions that require urgent surgical intervention such as type A AD. The triage system could provide real-time alerts for the clinicians and radiologists once aortic CTA was done. Future research for further validation of the model across different healthcare systems may be helpful for managing AD. Changes can be found on page 9.

Reviewer 3 Report

Comments and Suggestions for Authors

Hung-Hsien, Liu et al. present an original article with the aim to develop and validate a model system using deep learning algorithms for automatic detection of type A aortic dissection (AD) and differentiate it from normal and type B AD patients. The article is interesting, even if this is a retrospective study. Abstract and discussion should be improved, as well as the figures. Results are relevant for the field.

1. Abstract should be improved, it is very short. Add details about results.

2. Introduction should be improved by adding more information on “Deep learning”.

3. I suggest to add a figure showing the Stanford classification.

4. I suggest to improve the discussion by adding a paragraph on the application of your findings in clinical practice.

5. I suggest to add in the Discussion a paragraph on the “future research direction” and the possible application of the multimodality imaging in this setting and with this algorithm (AD and difference with the other acute aortic syndromes, for example). Please, add this key reference to improve the scientific content of the new sentence: “Perone F, et al. The Role of Multimodality Imaging Approach in Acute Aortic Syndromes: Diagnosis, Complications, and Clinical Management. Diagnostics (Basel). 2023 Feb 9;13(4):650. doi: 10.3390/diagnostics13040650”.

Author Response

Author's Reply to the Review Report (Reviewer 3)

Comment 1: Abstract should be improved, it is very short. Add details about results.

Response 1: Thank you for your kind suggestion. We’ve revised the abstract to add more details to the results section, emphasizing the component of our model. The changes could be seen on page 1, line 6-9.

Comment 2: Introduction should be improved by adding more information on “Deep learning”.

Response 2: Thank you for pointing this out. Therefore, we have added more information in the introduction section as follow: Deep learning models have been developed to detect AD with comparable diagnostic performance to radiologists, which showed the potential to support clinical practice (9-11). Raj et al created an algorithm that could detect AD in the abdomen (12). In contrast to their study, we intended to develop a deep learning model focusing on AD in the thorax, that would potentially be a very useful tool for automatic detection of the presence of AD for aortic CTA… The changes could be found on page 2, line 16-20.

Comment 3: I suggest to add a figure showing the Stanford classification.

Response 3:

hank you for kind suggestions. A sample image and corresponding figure legend were added in the dataset section. The changes could be found on page 3, section 2.2, line 11 in the revised manuscript and they were also demonstrated immediately below:

Figure 1. Example of aortic CTA in our study. Figure 1A showed aortic dissection (AD) in-volved the ascending and descending aorta, which was typical imaging presentation of Stanford type A AD. Figure 1B displayed an intimal flap in the descending aorta suggesting a Stanford type B AD. Figure 1C was a normal case.

Comment 4: I suggest to improve the discussion by adding a paragraph on the application of your findings in clinical practice.

Response 4: Thank you for your kind feedback. We’ve added a paragraph in the discussion section to illustrate the application of our model in clinical practice and future research direction as suggested in the next comment. The changes could be seen in the last paragraph of discussion on page 9, line 26-37.

Comment 5. I suggest to add in the Discussion a paragraph on the “future research direction” and the possible application of the multimodality imaging in this setting and with this algorithm (AD and difference with the other acute aortic syndromes, for example). Please, add this key reference to improve the scientific content of the new sentence: “Perone F, et al. The Role of Multimodality Imaging Approach in Acute Aortic Syndromes: Diagnosis, Complications, and Clinical Management. Diagnostics (Basel). 2023 Feb 9;13(4):650. doi: 10.3390/diagnostics13040650”.

Response 5: Thank you for your kind suggestion. We’ve revised the manuscript in the discussion section and added the key reference in the new sentences as recommended. The changes could be found in the last paragraph of discussion on page 9, line 26-37.

Reviewer 4 Report

Comments and Suggestions for Authors

A deep learning model has been proposed for the classification of type A aortic dissection (AD), normal and type B AD patients using computed tomography angiography (CTA) scans. The issues that need to be corrected in the article are as follows.

Minor:

Edit the reference formats in the text.

Summarize the contributions made to the literature at the end of the introduction section.

Add a sample image for normal and diseased images in the Dataset section.

Add the layers and features of the proposed model to the article as a table.

Major:

It is important to determine that the training process of deep learning models is completed. Add the ROC, loss function and accuracy changes during the training process to the article. Briefly comment on these changes. For example, if the training is completed, the changes should be minimal or stable after a certain epoch.

A total of 7 faulty data were classified in the confusion matrix. In fact, Normal and A classes should be examined within this data. There are a total of 4 (3+1) faulty data for these classes. It is a big problem to describe a healthy individual as a patient or a sick individual as a healthy individual. I suggest that this issue be detailed in the discussion section. Is it really difficult to distinguish these erroneous findings?

Why was k-cross validation not applied? K-cross validation could have been used to obtain more reliable results.

Author Response

Author's Reply to the Review Report (Reviewer 4)

Comment 1: Edit the reference formats in the text.

Response 1: Thank you very much. The reference formats were edited and changed into ACS style. The changes could be seen in the reference section from page 10-11.

Comment 2: Summarize the contributions made to the literature at the end of the introduction section.

Response 2: The contribution made to the literature was described in the author contributions section, which could be seen on page 10.

Comment 3: Add a sample image for normal and diseased images in the Dataset section.

Response 3: Thank you for kind suggestions. A sample image and corresponding figure legend were added in the dataset section. The changes could be found on page 3, section 2.2, line 11 in the revised manuscript and they were also demonstrated immediately below:

Figure 1. Example of aortic CTA in our study. Figure 1A showed aortic dissection (AD) in-volved the ascending and descending aorta, which was typical imaging presentation of Stanford type A AD. Figure 1B displayed an intimal flap in the descending aorta suggesting a Stanford type B AD. Figure 1C was a normal case.

Comment 4: Add the layers and features of the proposed model to the article as a table.

Response 4: Thank you for your suggestion. However, the models we utilized, RetinaNet and EfficientNet, are well-established architectures that have been employed in hundreds, if not thousands, of applications, and we did not make any modifications to their architectures. The most comprehensive descriptions of these models can be found in their original papers, references listed below (1-3). Additionally, previous works (4-5) also did not provide tables detailing the layers and features of the architectures they employed, as they used widely recognized models without modifications.

1) Lin, T.-Y.; Goyal, P.; Girshick, R.; He, K.; Dollar, P. Focal Loss for Dense Object Detection. Proc. IEEE Int. Conf. Comput. Vis. 2017, 2980–2988. DOI: 10.1109/ICCV.2017.324.

2) Tan, M.; Le, Q. V. EfficientNet: Rethinking Model Scaling for Convolutional Neural Networks. Proceedings of the 36th International Conference on Machine Learning, 2019, 6105–6114. DOI: 10.48550/arXiv.1905.11946

3).He, K.; Zhang, X.; Ren, S.; Sun, J. Deep Residual Learning for Image Recognition. Proceedings of the IEEE Conference on Computer Vision and Pattern Recognition, 2016, 770–778. DOI: 10.1109/CVPR.2016.90.

4) Hata, A.; Yanagawa, M.; Yamagata, K.; Suzuki, Y.; Kido, S.; Kawata, A.; Doi, S.; Yoshida, Y.; Miyata, T.; Tsubamoto, M.; et al. Deep learning algorithm for detection of aortic dissection on non-contrast-enhanced CT. Eur Radiol 2021, 31 (2), 1151-1159. DOI: 10.1007/s00330-020-07213-w  From NLM Medline.

5) Huang, L. T.; Tsai, Y. S.; Liou, C. F.; Lee, T. H.; Kuo, P. P.; Huang, H. S.; Wang, C. K. Automated Stanford classification of aortic dissection using a 2-step hierarchical neural network at computed tomography angiography. Eur Radiol 2022, 32 (4), 2277-2285. DOI: 10.1007/s00330-021-08370-2  From NLM Medline.

Comment 5: It is important to determine that the training process of deep learning models is completed. Add the ROC, loss function and accuracy changes during the training process to the article. Briefly comment on these changes. For example, if the training is completed, the changes should be minimal or stable after a certain epoch.

Response 5: We appreciate your feedback. The training logs are not saved due to limited storage resources. Model performance in terms of ROC, loss and accuracy throughout the training process would require all of our models to be retrained and would have been simply infeasible within ten days. However, we chose the object detection model which was trained with 1198 epochs and the classification model which was trained with 99 epochs. Training ends if training loss and validation loss is stable for at least 20 epochs. Moreover, none of the works related to these previous ones (1-4) reported the change of ROC, loss and accuracy during training.

  • Hata, A.; Yanagawa, M.; Yamagata, K.; Suzuki, Y.; Kido, S.; Kawata, A.; Doi, S.; Yoshida, Y.; Miyata, T.; Tsubamoto, M.; et al. Deep learning algorithm for detection of aortic dissection on non-contrast-enhanced CT. Eur Radiol 2021, 31 (2), 1151-1159. DOI: 10.1007/s00330-020-07213-w From NLM Medline.
  • Yi, Y.; Mao, L.; Wang, C.; Guo, Y.; Luo, X.; Jia, D.; Lei, Y.; Pan, J.; Li, J.; Li, S.; et al. Advanced Warning of Aortic Dissection on Non-Contrast CT: The Combination of Deep Learning and Morphological Characteristics. Front Cardiovasc Med 2021, 8, 762958. DOI: 10.3389/fcvm.2021.762958 From NLM PubMed-not-MEDLINE
  • Huang, L. T.; Tsai, Y. S.; Liou, C. F.; Lee, T. H.; Kuo, P. P.; Huang, H. S.; Wang, C. K. Automated Stanford classification of aortic dissection using a 2-step hierarchical neural network at computed tomography angiography. Eur Radiol 2022, 32 (4), 2277-2285. DOI: 10.1007/s00330-021-08370-2 From NLM Medline.
  • Harris, R. J.; Kim, S.; Lohr, J.; Towey, S.; Velichkovich, Z.; Kabachenko, T.; Driscoll, I.; Baker, B. Classification of Aortic Dissection and Rupture on Post-contrast CT Images Using a Convolutional Neural Network. J Digit Imaging 2019, 32 (6), 939-946. DOI: 10.1007/s10278-019-00281-5 From NLM Medline.

Comment 6: A total of 7 faulty data were classified in the confusion matrix. In fact, Normal and A classes should be examined within this data. There are a total of 4 (3+1) faulty data for these classes. It is a big problem to describe a healthy individual as a patient or a sick individual as a healthy individual. I suggest that this issue be detailed in the discussion section. Is it really difficult to distinguish these erroneous findings?

Response 6:

Thank you for your suggestion. We apologize for providing the incorrect version of the confusion matrix, which has now been replaced with the correct one. In the revised confusion matrix, no sick individuals are predicted as healthy. There was an error in the imaging itself and the case should have been excluded due to missing data. In the revised confusion matrix, the total number of sample sizes now matched the number of independent test set (n = 316). The revised confusion matrix was demonstrated immediately below and the changes could be seen in figure 4 on page 7

As demonstrated by the confusion matrix, 3 normal cases were interpretated as type A aortic dissection by the model. The reason for misinterpretation was due to steaking artifact, which was caused by dense contrast medium in the superior vena cava. The aortic CTA was generally performed by a fast injection rate of contrast medium so that the contrast medium could be accumulated in the superior vena cava. The artifacts could be minimized by using filtration, calibration correction, and beam hardening correction software. We’ve revised the manuscript, and changes could be found on page 8, line 19-20.

Comment 7: Why was k-cross validation not applied? K-cross validation could have been used to obtain more reliable results.

Response 7:

Thank you for your question. While k-fold cross-validation could provide a more reliable model evaluation, our system consists of two deep learning models designed for different tasks. Implementing k-fold cross-validation would complicate the process significantly and extend the training time for both models. Instead, we divided our dataset into three distinct sets: training, validation, and testing, which effectively demonstrates the generalizability of our object detection model. From the same 498 patients, we cropped all the labeled aorta areas into 31,622 patches, which were then randomly divided into the training, validation, and testing sets for the classification model. Additionally, we have a large independent test set containing 316 cases. We believe these process and evaluation are sufficient to establish the generalizability of our models. We have included a description of how to prepare our dataset for training the classification model in the first paragraph of Section 3.1 on page 5, along with Table 2 on page 6 to provide further details about our classification dataset.

Round 2

Reviewer 2 Report

Comments and Suggestions for Authors

Researchers have explained all the details that were wondered in the previous revision.

Reviewer 3 Report

Comments and Suggestions for Authors

The authors have responded satisfactorily to my requests

Reviewer 4 Report

Comments and Suggestions for Authors

The fact that ROC, loss, and accuracy changes were not given in previous studies should not be a valid reason for you. Your statement that the training process was completed is enough to convince me. However, I cannot say this for the researchers who read the article. Thanks for all the appropriate responses to the comments and additions to the article.